# Oxalic Acid Inhibits Feeding Behavior of the Brown Planthopper via Binding to Gustatory Receptor Gr23a

**DOI:** 10.3390/cells12050771

**Published:** 2023-02-28

**Authors:** Kui Kang, Mengyi Zhang, Lei Yue, Weiwen Chen, Yangshuo Dai, Kai Lin, Kai Liu, Jun Lv, Zhanwen Guan, Shi Xiao, Wenqing Zhang

**Affiliations:** 1State Key Laboratory of Biocontrol and School of Life Sciences, Sun Yat-sen University, Guangzhou 510275, China; 2School of Biology and Agriculture, Zunyi Normal University, Zunyi 563006, China

**Keywords:** ligand identification, gustatory receptor, *Nilaparvata lugens*, oxalic acid, antifeedant

## Abstract

Plants produce diverse secondary compounds as natural protection against microbial and insect attack. Most of these compounds, including bitters and acids, are sensed by insect gustatory receptors (Grs). Although some organic acids are attractive at low or moderate levels, most acidic compounds are potentially toxic to insects and repress food consumption at high concentrations. At present, the majority of the reported sour receptors function in appetitive behaviors rather than aversive taste responses. Here, using two different heterologous expression systems, the insect *Sf9* cell line and the mammalian HEK293T cell line, we started from crude extracts of rice (*Oryza sativa*) and successfully identified oxalic acid (OA) as a ligand of *Nl*Gr23a, a Gr in the brown planthopper *Nilaparvata lugens* that feeds solely on rice. The antifeedant effect of OA on the brown planthopper was dose dependent, and *NlGr23a* mediated the repulsive responses to OA in both rice plants and artificial diets. To our knowledge, OA is the first identified ligand of Grs starting from plant crude extracts. These findings on rice–planthopper interactions will be of broad interest for pest control in agriculture and also for better understanding of how insects select host plants.

## 1. Introduction

Plant visual and chemical cues are important for host location and host acceptance of phytophagous insects. The cabbage root fly, *Delia radicum*, makes use of leaf colors to discriminate among host plants [1]. Host selection in oligophagous species is involved with the balance of phagostimulatory and deterrent inputs with the addition of a specific chemical sign stimulus [2]. Understanding the mechanisms underlying this selection strategy is of practical importance in agriculture for pest control. For instance, tandem deployment of the napier grass, which attracts greater oviposition by stemborer moths than maize but is not suitable for survival, and the molasses grass, which causes over 80% reduction in stemborer infestation of maize, can interfere with host selection by stemborers [3]. The combination of stimuli that have negative effects on food selection by pests and positive effects on diverting pests from the protected resource to a trap can reduce pest abundance on the protected resource; this is called the push–pull strategy [4]. Plant-derived antifeedants can be used as push components of the push–pull strategy [4]. Identification of receptors sensing these chemicals will definitely improve our understanding of how insects select host plants.

To date, most identified antifeedants are secondary metabolites, which are toxic or deterrent to insects [2,5,6]. As an example, oxalic acid (OA), the simplest dicarboxylic acid common in plant tissues, provides protection for plants against insects and foraging animals, including the brown planthopper (BPH), *Nilaparvata lugens,* and aphids [6,7,8]. Some antifeedants (e.g., tricin) could be even used as indicators of crop resistance to target insect pests, and are therefore potentially valuable for breeding resistant crop varieties [9].

In insects, gustatory receptor neurons (GRNs), mainly found in gustatory sensilla, are able to convert the chemical signal into an electrical one and transmit it to higher-order brain structures for processing, which in turn dictates behavior [10]. Secondary compounds as deterrents stimulate a subset of bitter GRNs, which then inhibit feeding activity or induce repulsion [2]. With the decoding and publication of the genomes of *Drosophila melanogaster* and other insect species, fruitful molecular studies on crop pest gustation have been enabled. Gustatory receptors (Grs), expressed in GRNs, are required for responses to specific tastants. Grs mainly include sweet, bitter, and CO_2_ receptors [11]. Although diverse sugars and bitter compounds have been identified as ligands of insect Grs, many Grs have yet to be functionally annotated, particularly those sensing acids [12]. In *D. melanogaster*, sweet gustatory receptors were reported to non-specifically respond to low pH with mild attraction [13,14]. The fly also uses ionotropic receptors and mammalian orthologs to perceive acids [12,15,16]. However, there is currently no report of insect Grs sensing aversive acids derived from plants.

The BPH, causing extensive damage by sap feeding and transmitting viruses, feeds solely on *Oryza sativa* and its allied wild forms, such as *O*. *perennis* and *O. spontanea* [17]. Its gustatory sensilla are mainly located in the small passageway leading from the food duct to the cibarium and the stylet groove on the labial tip, and probably regulate the initial stages of probing and the maintenance of sap ingestion [18,19]. In our previous study, 32 *N. lugens Gr* sequences (*NlGrs*) were obtained following analyses of the BPH genome and transcriptomes [20]. In previous studies, we explored the roles of *Nl*Grs sharing orthologous relationships with *D. melanogaster* Grs [21,22,23]. Here, we would like to investigate *Nl*Grs with unknown functions. In insects, most functionally characterized Grs have seven transmembrane (TM) domains. However, only a few Grs are 7-TM proteins in the BPH. *Nl*Gr23a, which was predicted to possess typical GR characteristics with 7-TM domains and was successfully cloned, was selected as the focus of the present study. Through a multi-stage bioassay-directed fractionation of rice crude extracts, we identified OA as a ligand of *Nl*Gr23a. We also found that *NlGr23a* mediated the repulsive responses to OA in both rice plants and artificial diets.

## 2. Material and Methods

### 2.1. Insect and Rice Culture

A BPH laboratory strain (*Ctl*) was obtained from Guangdong Academy of Agricultural Sciences (GDAAS, Guangdong, China), and this strain was reared in a continuous laboratory culture on susceptible rice seedlings (variety Huang Hua Zhan). All BPHs were maintained at 26 ± 2 °C with 80 ± 10% humidity and a light–dark cycle (16 h light and 8 h dark). One-day-old brachypterous female adults and the susceptible rice variety Taichung Native 1 (TN1) aged 30–40 days were used for the following bioassays. Allocation of insects was randomly done to minimize the effects of subjective bias.

### 2.2. Chemical Sources

OA, glycerine, adonitol, methoxylamine hydrochloride, N-methyl-N-(trimethylsilyl) trifluoroacetamide, tetrabutylammonium bisulphate (TBA), KH_2_PO_4_, and caffeine (purity, ≥99%) were purchased from Sigma-Aldrich Company (MO, USA). HPLC grade reagents including petroleum ether (PE), chloroform (CH_3_Cl), ethyl acetate (EAC), methyl alcohol (MeOH), acetonitrile (ACN), isopropyl alcohol, and pyridine, as well as analytical grade reagents including sucrose, trans-aconitic acid, salicylic acid, mandelic acid, and ethyl alcohol were obtained from Aladdin Reagent (Shanghai, China). Maleic acid was purchased from MedChemExpress (Shanghai, China). Hydrochloric acid (HCl) was provided by the Guangzhou Chemical Reagent Factory (Guangzhou, China).

### 2.3. Ligand Determination for NlGr23a

#### 2.3.1. *NlGr23a* cDNA Cloning and Cell Line Construction

Total RNA from BPHs was prepared as previously described [20]. A PrimeScript^TM^ RT reagent kit with gDNA Eraser (Takara, Kyoto, Japan) was used with 1 µg RNA for first-strand complementary DNA (cDNA) synthesis. A fragment of full-length *NlGr23a* cDNA was amplified using 2 × SuperStar PCR Mix with Loading Dye (Genstar, Beijing, China) following the manufacturer’ss instructions. The cloning primers are listed in Appendix A. The *NlGr23a* cDNA was inserted into the Hind III and EcoR I sites of pIZ/V5-His vectors (Invitrogen, Carlsbad, CA, USA) using T4 DNA ligase (NEB, Beijing, China). The *Nl*Gr23a-*Sf9* stable cell lines were obtained according to a previously described method [21]. Briefly, *Sf9* cells were cultured in Grace’s Insect Medium (Gibco, Grand Island, NY, USA) supplemented with 10% fetal bovine serum at 27 °C. Cells were plated into 6-well plates and left to settle for 20 min before being transfected with either 2 µg of the recombinant plasmid pIZ-*Nl*Gr23a-V5-His or pIZ-V5-His vector (negative control), and 6 µL Fugene HD transfection reagent (Promega, Madison, WI, USA) in 100 µL per well of Grace’ss Insect Medium. Forty-eight hours after transfection, cells were cultivated under Zeocin selection. The final concentration of Zeocin to maintain cells was 100 µg/mL to obtain stable cell lines. To further confirm the sufficiency of *Nl*Gr23a for OA-mediated response, the HEK293T cells were transiently transfected with *Nl*Gr23a/pcDNA3.1-FLAG. The HEK293T cells were cultured in Dulbecco’ss modified Eagle medium (Gibco, CA, USA) supplemented with 10% fetal bovine serum in an atmosphere of 5% CO_2_ and 95% relative humidity at 37 °C. *Nl*Gr23a/pcDNA3.1-FLAG was constructed by inserting a C-terminally FLAG-tagged human codon-optimized *NlGr23a* ORF into the pcDNA3.1 vector using BamHI and EcoRI sites. The nucleotide sequences of the codon-optimized genes were listed in Appendix A. The HEK293T cells were transfected with 0.5 µg plasmid DNA per well using 1.5 µL Lipofectamine^TM^ 3000 Reagent and 1 µL P3000™ Reagent (both from Invitrogen, Carlsbad, CA, USA), following the manufacturer’s protocol.

#### 2.3.2. Isolation and Identification of Active Compounds in Rice

A calcium imaging assay guided approach was adopted to isolate and identify active compounds interacting with *Nl*Gr23a from the crude extracts of rice plants. Firstly, we ground stems and leaves of TN1 (5 g) and performed serial solvent extractions on macerated subsamples with 25 mL of isopropanol (IPA) at 75 °C for 15 min, 25 mL CHCl_3_-IPA (1:2, *v*/*v*; 5 h; room temperature), and 25 mL MeOH-CHCl_3_ (2:1, *v*/*v*; overnight; room temperature). Each extract was centrifuged at 2000 rpm for 15 min to collect the supernatant. The pooled supernatants were then concentrated to dryness in vacuum at 25 °C and re-dissolved in 12 mL CHCl_3_ to obtain the crude rice extract. Secondly, 6 mL crude extract was subjected to a column chromatography with silica gel, eluting with solvents in the order of increasing polarity (PE-CHCl_3_-EAC-MeOH) to yield four fractions. We carried out the first round of ligand screening with four initial fractions. Thirdly, semi-preparative HPLC separation of the bioassay-active fraction (active fraction I) of the first round screening was performed on an Agilent 1260 HPLC system (Agilent Technologies, Santa Clara, CA, USA) equipped with an Agilent ZRBAX SB-C18 column (4.6 mm × 150 mm, 3.5 µm) by gradient elution with ACN (A) and water (B). The gradient program was as follows: 3 min, 5% B; 22 min, 5–20% B; 15 min, 20–40% B; 15 min, 40–50% B; 8 min, 50–95% B; 12 min, 95–5% B, all at a flow-rate of 1 mL/min. The eluate from the column was continuously monitored at 205 and 290 nm after sample loading (5 µL), and the fraction collector was programmed to collect fractions during periods in which most substances were detected. The second round of screening for the ligand was then carried out with these subfractions. Fourthly, the active semi-prep-HPLC fraction (active fraction II) was subjected to further fractionation with 100% ACN at a flow-rate of 1 mL/min for the third round of screening. Last, the bioassay-positive fraction (active fraction III) from the third round was derivatized and elucidated using GC-MS. The GC-MS detection for the derived samples was performed on an Agilent 7890A gas chromatograph equipped with an Agilent 5975C VL MSD detector (Agilent Technologies, Santa Clara, CA, USA) as previously described [24].

### 2.4. Calcium Imaging Assay

Calcium imaging using calcium indicator dye was performed as previously described, with some modifications [21,25]. Cells were seeded in a glass bottom cell culture dish (*φ* 20 mm, Nest, Jiangsu, China) at 70% confluency and grown overnight. After being washed in Hanks’ balanced salt solution (HBSS, Solarbio, Beijing, China) three times, cells were incubated for 30 min with 2 mL HBSS, which was added with 1 µM Fura2-AM or 5 µM Fluo4-AM (Invitrogen, Carlsbad, CA, USA) under shaking conditions (60 rpm) at room temperature in darkness. Subsequently, calcium indicator dye was removed, and cells were then washed twice with HBSS and were covered with 2 mL of fresh HBSS. Imaging experiments were conducted on a Leica DMI6000B confocal microscope (Leica Microsystems, Wetzlar, Germany) equipped with a CCD camera. *Sf9* cells transfected with Fura2-AM were stimulated with 340 and 380 nm, and emission was set at 510 nm. HEK293T cells transfected with Fluo4-AM were stimulated with 488 nm, and emission was set at 518 nm. A total of 200 µL test solution was added into the dish using a pipette. Data acquisition and analysis were performed using Leica LAS-AF software (version 2.6.0). Cell assays were repeated for at least three times. Sample size of cell assays to achieve adequate power was chosen on the basis of a previous report [26].

### 2.5. RNAi Treatments

#### 2.5.1. dsRNA Preparation and Injection into the BPH

Template DNA for dsRNA synthesis was amplified using gene-specific primers (see Appendix A). The resulting purified products were then used to synthesize dsRNA using a MEGAscript T7 High Yield Transcription Kit (Promega, Madison, WI, USA). The concentration of *NlGr23a* dsRNA (239 bp) was quantified with a NanoDrop 2000 instrument (Thermo Fisher Scientific, Waltham, MA, USA). Finally, the quality and size of the dsRNA were further verified via electrophoresis in a 1% agarose gel. *GFP* dsRNA was used as a control. BPHs were collected from the culture chamber and anesthetized with CO_2_ for 20 s. Approximately 250 ng dsRNA was injected into each individual. After injection, BPHs were reared on fresh rice plants. Individuals exhibiting motor dysfunction and/or paralysis in 12 h were excluded from next steps. Total RNA of the whole body was extracted from five randomly selected individuals to examine the gene silencing efficiency at different time-points (24  h, 48  h and 72  h post injection) by quantitative real-time PCR (qRT-PCR). Bioassays were conducted 48 h after injection.

#### 2.5.2. qRT-PCR Analysis

The qRT-PCR procedure was performed using a Light Cycler 480 (Roche Diagnostics, Basel, Switzerland) with a SYBR^®^ FAST Universal qPCR Kit (KAPA, Woburn, MA, USA), following the manufacturer’s instructions. Each reaction mixture included 1 µL of cDNA template equivalent to 1 ng of total RNA, 0.3 µL of each primer (10 µM), and 5 µL SYBR mix in a total volume of 10 µL. The experiment was repeated for three biological replicates, and three reactions for each biological replicate were performed. Gene expression levels were normalized to the expression level of BPH *β*-*actin* [27]. The specific primers used for qRT-PCR are listed in Appendix A. The amplification conditions were as follows: 95 °C for 5 min, followed by 45 cycles of 95 °C for 10 s, 60 °C for 20 s, and 72 °C for 20 s.

### 2.6. Behavioral Experiments

#### 2.6.1. Feeding Bioassays of BPHs on Rice Plants

Rice with a higher content of OA (TN1^+OA^) was obtained by the immersion method. Briefly, TN1 rice at the tillering stage was immersed in OA solution (20 mM) for 5 min and air-dried for 1 h. The resulting OA level was determined as follows. Firstly, the rice was washed and drained, then ground to a powder using liquid nitrogen. Secondly, the rice powder was placed in a 50 mL centrifuge tube and mixed with 0.5 mol/L HCl (fresh weight of rice:HCl = 1:5; *w*:*v*). The homogenate was placed in a boiling water bath for 15–20 min and then centrifuged at maximum speed. The resulting supernatant was diluted with ddH_2_O. Finally, the sample was filtered with a microporous filter membrane (0.45 µm). The OA content was determined using an UltiMate 3000 HPLC (Dionex Corporation, Sunnyvale, CA, USA) equipped with an Agilent ZRBAX SB-C18 column (4.6 mm × 150 mm, 3.5 µm). The mobile phase was 5 mM TBA in 0.5% KH2PO4 (pH 2.0) with a flow rate of 1 mL/min. The eluate from the column was continuously monitored at 220 nm after sample loading (5 µL).

To evaluate BPHs food preferences across rice diets with different OA levels, we performed two-choice assays. At the beginning of each experiment, ten BPHs were placed on the middle of a sponge, which sealed an upside-down plastic cup containing one TN1 stem and one TN1^+OA^ stem. Insects were starved for 6 h before each experiment. We scored the distribution of insects on TN1 or TN1^+OA^ at 30 min intervals for 4 h. Values were totaled, and a position index (PI) for the 4 h period was calculated. PI = (BPHs on TN1^+OA^ − BPHs on TN1)/(BPHs on TN1^+OA^ + BPHs on TN1). The PI values ranged from −1 to 1, with PI = 0 indicating no position preference for either TN1 or TN1^+OA^, PI > 0 indicating preference for TN1^+OA^, and PI < 0 indicating preference for TN1. The experiment was repeated with five biological replicates. Probing marks were detected based on a previously described method [28]. Briefly, BPHs (*n* = 4) reared on one rice stem were confined for 4 h using a 50 mL polypropylene tube. The exposed plant parts were stained with 1% eosin Y (Aladdin Reagent, Shanghai, China), then probing marks on the plant surface were counted. The experiment was repeated with five biological replicates.

*OsICL* over-expression lines were developed by Towin Biotechnology Co., Ltd (Wuhan, China) in accordance with a previous report [29]. The rice (*O. sativa* ssp. *japonica*) wild-type (WT) and transgenic plants used in this study were in the cv Nipponbare background. T2 transgenic lines resistant to hygromycin were chosen for the analysis of the OA content. For honeydew excretion assay, female adults were enclosed in a pre-weighed parafilm sachet (each parafilm sachet contains three female adults) that was attached to the leaf sheath of the rice plant. The honeydew of each parafilm sachet was weighed after 48 h. The weight change of the sachet was recorded as the honeydew excretion. Twelve replications were carried out. For host choice assay, two *OsICL*ox transgenic plants and two WT seedlings placed alternately were confined in ventilated plastic cylinders and infested with 30 females. The numbers of BPHs settling on each plant were counted at 48 h post release. Twelve replications were carried out.

#### 2.6.2. Feeding Bioassays of BPHs on Artificial Diets

The no-choice test apparatus for evaluation of feeding decision behavior contained a glass tube, parafilm, and artificial diet D-97 [30]. Two layers of stretched parafilm encased the diet (40 µL), either with OA (100 μM, +OA) or without (−OA), and sealed the tube. BPHs (*n* = 6–10) were introduced into the tube placed onto the food-containing parafilm at the beginning of the experiment in each of three replicates. We counted the insects remaining on each parafilm (N_on_) at 30 min intervals for 4 h. The proportion of BPHs accepting food sources was calculated as food acceptance (food acceptance = N_on_/N_total_), where N_total_ represents the total number of test insects. To evaluate the position preference of BPHs between two ends with artificial diets with or without OA, we performed dual choice assays. Parafilms containing 40 µL of either +OA or −OA artificial diet were located at the two ends of a glass tube separately. Ten BPHs per replicate were introduced into the middle of the cylinder at the beginning of the experiment; four replicates were performed.

In the EPG recording assay, BPH feeding behavior was recorded on a Giga-8 DC EPG amplifier (GDAAS, Guangdong, China). All experiments were carried out at 26 °C ± 1 °C and 70% ± 10% relative humidity under continuous light conditions. The feeding behavior of individual BPH on liquid diet sacs (LDS) was monitored for 3 h. Each treatment was replicated 5 times. The signals recorded were analyzed using PROBE 3.4 software (Wageningen Agricultural University, Wageningen, The Netherlands). Each feeding behavior was expressed as the duration of each waveform as a proportion of total monitoring time (%).

To directly assess the feeding behavior, we performed food choice assays as described previously, with modifications [31,32]. Briefly, 10 BPHs were starved for 6 h and introduced into a glass tube. Then, two layers of parafilm wrapping 2% or 10% sucrose solution sealed the tube. The indicated concentration of OA was diluted with 10% sucrose solution and colored with blue dye (Brilliant Blue FCF, 0.2 mg/mL) and 2% sucrose solution was mixed with red dye (sulforhodamine B, 0.1 mg/mL). BPHs were allowed to make a choice between these two sucrose solutions for 4 h. After feeding, BPHs were dissected to observe their mid-gut colors under the microscope for the presence of red (N_R_), blue (N_B_), or purple (N_P_) dye. The preference index was calculated using the following equation: preference index = (N_R_ − N_B_)/(N_R_ + N_B_ + N_P_). A preference index of −1.0 or 1.0 indicated a complete preference for either 10% sucrose with OA or 2% sucrose alone, respectively. A preference index of 0 indicated no bias between the two food alternatives. Each treatment had at least eight biological repeats.

### 2.7. Surgeries

To investigate whether olfaction was involved with OA sensing in BPHs, female insects were anesthetized with CO_2_. Then, the second and third antennal segments, which are the primary olfactory organs of the BPH [33], were removed using a spring scissor. Finally, food choice assays with antennectomized insects were performed as before.

### 2.8. Immunohistochemistry and Microscopy

The tissue slices were processed for immunofluorescence microscopy as previously described [34]. The labium of a one-day-old brachypterous female adult was removed, washed with 70% ethanol, and fixed in 4% paraformaldehyde at room temperature for 2 h. After fixation, the labium was embedded in Tissue-Tek O.C.T. compound (Sakura Finetek, Tokyo, Japan) and frozen at −20 °C. The embedded specimens were mounted on an object holder and cryosectioned using a Leica CM1950 cryostat (Leica Biosystems, Wetzlar, Germany) at 6 µm thickness. The cryosections were placed on adhesion glass slides (CITOGLAS, Jiangsu, China) and air dried at room temperature for 4 h. HPLC purified rabbit antibody against *Nl*Gr23a peptide CTLESRKVLSIKSKN (8 µg/mL) was used as the primary antibody, and Alexa Fluor 555^®^-conjugated goat anti-rabbit antibody (1:400; Invitrogen, Carlsbad, CA, USA) was used as the secondary antibody. Prior to observation, the samples were stained with Hoechst 33342 (Invitrogen, Carlsbad, CA, USA) and washed two times with PBS. The tissue slices were visualized with a Zeiss LSM 880 laser scanning confocal microscope (Carl Zeiss, Oberkochen, Germany).

### 2.9. Statistics

All statistical analyses were performed using IBM SPSS Statistics 24 (IBM, Armonk, NY, USA). The non-parametric Mann–Whitney *U* test was used to test for significant differences between two groups. Comparisons within multiple groups were evaluated with one-way ANOVA followed by Duncan’s test. Data were checked for normal distribution using the Shapiro–Wilk test.

## 3. Results

### 3.1. Oxalic Acid Is a Ligand of NlGr23a

To identify the specific ligand of *Nl*Gr23a, we cloned the full-length *NlGr23a* open reading frame (GenBank accession No. MT387198; date last accessed: Feb 2023), which encodes 451 amino acid residues with seven predicted transmembrane domains (Appendix A). Then, the *Nl*Gr23a-expressing stable *Spodoptera frugiperda* (*Sf9*) cell line was established to carry out four rounds of ligand screening from crude extracts of rice stems and leaves. The initial calcium imaging results revealed that the ethyl acetate (EAC) and methyl alcohol (MeOH) fractions caused significantly higher calcium ion (Ca^2+^) concentration (i.e., cellular response) as indicated by the ratio of cytoplasmic fluorescence intensities (Figure 1A). The EAC fraction was then further fractionated into three portions for the next round of screening; *Sf9* cells expressing *Nl*Gr23a showed a response only to fraction 2 (time of retention, *t_R_ =* 20–35 min, Figure 1B; see also Appendix A for HPLC chromatogram of the EAC fraction). This fraction was subjected to further fractionation with acetonitrile (ACN), and we found that only the subfraction collected in 5–10 min specifically evoked Ca^2+^ release (Figure 1C; see also Appendix A for HPLC chromatogram of the subfraction). The bioassay-positive subfraction was isolated and analyzed using gas chromatography-mass spectrometry (GC-MS). Four compounds, oxalic acid (OA), glycerol, phthalic acid, and trisiloxane, were identified (Figure 1D; see also Appendix A for GC-MS chromatogram of the control solution). As the peak intensity of phthalic acid was quite weak, and trisiloxane was probably an artefact of culture vessel silanization (see Appendix A for GC-MS analysis details), OA and glycerol were selected as potential ligands. Tests using commercial OA and glycerol showed that *Nl*Gr23a-*Sf9* cells responded dramatically to OA stimulation (Figure 1E). The responses increased with higher concentrations of OA solution (Figure 1F).

In addition, expressing *Nl*Gr23a in the human embryonic kidney 293T (HEK293T) cells also allowed these cells to respond to OA (Figure 1G). This suggested that *Nl*Gr23a showed full functionality in HEK293T cells without an insect-specific co-receptor. Given *Nl*Gr23a interacted with bioactive molecules in the MeOH fraction, *Nl*Gr23a may also respond to other ligands in rice (Figure 1A). To characterize the response profiles of *Nl*Gr23a, we tested responsiveness of the *Nl*Gr23a-expressing HEK293T cells to additional phytochemicals. Stimulation with only OA induced a Ca^2+^ increase in *Nl*Gr23a-HEK293T cells, whereas other organic acids showing inhibitory effects against BPHs elicited no response [6] (Appendix A). No response was also observed in sucrose, caffeine, and HCl (Appendix A). Hence, *Nl*Gr23a may be specifically required for OA sensing in a structure- and dose-dependent manner.

### 3.2. NlGr23a Is Required for Perception of Oxalic Acid in Rice Plants

We used the immersion method to increase the OA content of the rice variety TN1, resulting in 40% higher content in the OA-treated rice plants (TN1^+OA^) than in the untreated controls (Figure 2A). Insects fed TN1^+OA^ showed a positional avoidance response (position index, PI = −0.24) and made twice as many probing marks (caused by insects testing food for palatability and withdrawing) as those fed TN1, indicating that BPHs spent more time in searching for food on TN1^+OA^ (Figure 2B,D). Our analysis indicated that BPH feeding was inhibited as the content of OA in rice plants increased.

As OA was a ligand of the *Nl*Gr23a receptor, we hypothesized that the antifeedant activity of OA is dependent on this receptor in insects. The *NlGr23a* gene was therefore silenced using RNA interference (RNAi) in vivo. *NlGr23a* expression in the whole insect was decreased significantly 24 to 72 h post-injection (Appendix A). As expected, the positional avoidance response to OA decreased in ds*NlGr23a*-treated BPHs (PI = 0.02) but not in ds*GFP*-treated insects (PI = −0.23) (Figure 2C). In addition, ds*GFP*-treated BPHs produced 77% more probing marks on TN1^+OA^ rice plants than on TN1, but this difference disappeared after injection of *NlGr23a* dsRNA (Figure 2E). 

In rice, isocitrate lyase (ICL) catalyzes the production of glyoxylate, an efficient precursor for OA biosynthesis, by splitting isocitrate [29]. We used *OsICL*-overexpressing transgenic plants (*OsICL*ox) to stimulate OA accumulation. The OA level was significantly increased in *OsICL*ox (See Appendix A for OA contents in *OsICL*ox and WT plants). Compared with ds*GFP*-treated BPHs, which fed on transgenic rice plants, ds*NlGr23a*-treated BPHs feeding on *OsICL*ox plants showed a 26% increase in the weight of honeydew secretion (Figure 2F). Moreover, the interference of *NlGr23a* expression significantly increased the number of BPHs infesting transgenic plants (Figure 2G). These results suggested that *NlGr23a* mediated OA perception in rice. Then, we measured OA contents in three rice varieties and found there is a significant difference in OA content between the susceptible TN1 rice and the resistant rice varieties IR36 and IR56 (Appendix A). A population of sensory neurons tuned to OA is probably involved in detection of the dynamic changes of soluble OA in rice plants to provide clues for the suitable habitats or feeding times.

### 3.3. NlGr23a Mediated the Antifeedant Activity of Oxalic Acid in Artificial Diets

BPH adults offered artificial diets either containing (+OA) or missing OA (−OA) with no alternative (“no-choice” tests) showed significantly lower acceptance of +OA compared to −OA after 1.5 h exposure (Figure 3A). When offered a choice (“dual-choice”), +OA food was avoided in favor of −OA food, based on insect positioning relative to the food sources (PI = −0.52, Figure 3B). There were three main types of EPG waveforms that occurred during the process of BPH feeding on LDS: NP (non-penetration), PW (pathway wave), and N4 (artificial diet ingestion) (Figure 3C, top). OA, even at a low concentration, was thus capable of interfering with food sucking behavior. At 100 µM OA in the diet (+OA), the pre-feeding phases were longer than those of the control (−OA), and the N4 phase was very significantly shorter (Figure 3C, bottom). The proportion of time spent feeding fell further as OA concentrations increased, down to only 0.56% at an OA concentration of 10 mM (Figure 3C, bottom).

Further, we added two different food dyes into sucrose solutions with or without OA to measure direct feeding. BPHs were given a choice between 2% (*w*/*v*) sucrose and 10% sucrose plus different concentrations of OA. Sucrose is a potent sucking stimulant for BPHs [17]. In the absence of OA, the insects showed a preference for the five-fold higher concentration of sucrose (Figure 3D). However, when sucrose was contaminated with OA, BPHs avoided the OA-laced food in a dose-dependent manner (Figure 3D). A similar experiment was conducted using an equal concentration of sucrose (10%); BPHs showed similar preference for 10% sucrose solutions added with different food dyes without OA and exhibited repulsion to the food when sucrose solutions were mixed with OA (Appendix A). These results indicated that the antifeedant activity of OA was dose dependent. To exclude the possibility of olfaction-mediated OA sensing, we surgically removed the primary olfactory organs of the BPH, the second and third antennal segments [33]. Antennaectomized insects had normal OA avoidance (Appendix A). Thus, the olfactory system is not required for OA avoidance.

The antifeedant activity of OA was decreased in *NlGr23a*-inactive insects compared to the control insects injected with *GFP* dsRNA in no-choice tests. In BPHs fed −OA artificial diets, the level of food acceptance was nearly the same for both treated (ds*NlGr23a*) and control (ds*GFP*) groups (Figure 3E, top). In insects fed +OA diets, those with silenced *NlGr23a* spent more time on the OA-containing food than controls, significantly so between 1.5 h and 2.5 h after exposure (Figure 3E, bottom). In the dual-choice assays, BPHs injected with ds*NlGr23a* were less sensitive to OA (PI = −0.28) than controls (PI = −0.48, Figure 3F). When BPHs were fed on diets without OA, the NP duration was higher, and the N4 duration was commensurately lower in *NlGr23a*-silenced insects (Figure 3G, left), which indicated that RNAi against *NlGr23a* influenced their feeding behavior. At 10 mM OA, *NlGr23a*-silenced BPHs exhibited significantly more N4 and PW phases and fewer NP phases than controls (Figure 3G, right). Similar results were observed at 2 mM or 5 mM OA (Appendix A). Additionally, knockdown of *NlGr23a* dramatically reduced OA feeding avoidance (Figure 3H). Taken together, these results indicated that *NlGr23a*-silenced BPHs were less sensitive to OA in both rice plants and artificial foods.

### 3.4. Localization of NlGr23a in the Labium

The BPH has a highly modified labium adapted to a piercing and sucking method of feeding. To validate the expression of *Nl*Gr23a in the labium to sense OA, we performed immunohistochemistry analysis with the anti-*Nl*Gr23a antiserum to determine the distribution of *Nl*Gr23a-expressing cells in the oral sensory organs of female adults. The *Nl*Gr23a antibody labeled *Nl*Gr23a in the labial tip (Figure 4A–D). Negative control experiments were conducted using pre-immune, and no immunoreactive cells were observed (Figure 4E). Uniporous chemosensilla and domed multiporous chemosensilla were both present on the flattened labial tip, which is the first component of the mouthparts to touch the feeding substrate [18]. BPH individuals test the plant surface prior to probing by dabbing it with the labial tip [18]. *Nl*Gr23a found in the labial tip probably provides BPHs with information, which influences their subsequent feeding behaviors.

## 4. Discussion

OA is commonly present in rice plants. The species of only 11 out of 93 orders of higher plants do not store OA [35]. The fact that a secondary compound is deterrent to an insect does not imply that the insect does not eat plants containing that compound. Whether or not it does so depends on its sensitivity to the compound, the background of other deterrent and phagostimulatory information in which it is perceived, and the degree of food deprivation incurred by the insect [2]. Rice plants contain many metabolites, including both deterrents and stimulants to feeding [6,36]. In the EPG experiments of this study, 10 mM OA in artificial diets was capable of completely inhibiting feeding phases (Figure 3C). However, the OA content of the background material TN1 was measured as above 5 mM (Figure 2A). The difference in active OA concentration may be due to the different background of other chemicals. Moreover, OA concentrations in rice plants are dynamic, maybe varied in tissues, and affected by rhythm, rice variety, and developmental stages, as well as abiotic factors [37]. OA isolated from leaf sheath extracts of rice seems to mediate the phloem-feeding habit as a general inhibitor, which commonly occurs in plant tissues outside the phloem [6]. Such factors may also explain why the PI values in dual choice tests in rice tended to be higher than those from tests using the artificial diet containing OA (Figure 2B and Figure 3B). There are certainly other possible explanations. Firstly, in the case of the artificial diets, the diets either included or completely excluded OA, while in the case of rice plants, the OA content in TN1^+OA^ was only 39% higher than in TN1 (Figure 2A). Secondly, BPHs tend to move up and down along the same plant instead of quickly moving to a different plant.

Although insect bitter Grs are assumed to detect plant secondary compounds or bitter tastants, it is still very difficult to identify the ligands of bitter Grs, especially new examples. The pioneering identifications typically used electrophysiological and behavioral genetic analyses in *Gr* mutants of the model insect *D. melanogaster*. Additionally, most ligands tested to date have been directly purchased compounds, such as caffeine, strychnine, umbelliferone, saponin, chloroquine, and l-canavanine [31,38,39,40,41]. In other species, especially in agricultural insect pests, Grs are often assumed to match those of the orthologous fruit fly receptors. As non-sugar *Gr* genes are subject to rapid adaptation driven by the vastly different ecological niches occupied by insect species, and shared sequence identity among species is low, it is difficult to identify matching receptors. Alternatively, candidate ligands have been chosen from chemical classes that are known to elicit neurophysiological or behavioral responses. For example, a collection of known oviposition stimulants of *Papilionidae xuthus* were selected to test for interaction with *Pxut*Gr1, and eventually synephrine was identified as the specific ligand [42]. In order to expand potential ligand sources for non-sugar Grs, plant extracts have been tested in few cases. Extracts of whole citrus caused an increase in Ca^2+^-dependent luminescence in *Spodoptera frugiperda* 9 cells after the introduction of *PxutGr1* [42], and similar results were observed for crude extracts of cotton leaves and 3 *Grs* in *Helicoverpa armigera* (*HarmGr35*, *HarmGr50*, and *HarmGr195*) [43]. However, no specific ligand has so far been identified based on separation of these crude extracts. In this study, we started from rice crude extracts and eventually identified OA as a ligand of *Nl*Gr23a.

Insects discriminate a wide range of tastants, including sugars, bitter compounds, NaCl, and sour substances. In contrast to sweet and bitter-tasting chemicals, acids elicit varied behavioral responses depending on their structure and concentration [44]. To date, multiple classes of receptors have been reported to be involved in acid taste in *Drosophila*. For appetitive responses, Ir25a and Ir76b function in the legs for sensing proton concentration and the structure of carboxylic acids to mediate oviposition preference for acidic food [45]. Ir25a and sweet Grs are required for attractive taste sensation of lactic acid [14]. Multiple receptors, including Ir25a, Ir76b, Ir56d, and glycerol receptor Gr64e, contribute to the fatty acid recognition [13,46]. For aversive responses, Ir7a expressed in bitter GRNs is necessary for rejecting foods laced with high concentrations of acetic acid [44]. Moreover, Otopetrin-like A (OtopLA) forms a proton-conducting ion channel to transduct sour-taste and then mediates both the strong repulsion to highly acidic food and mild attraction to low acidity [15,16] In this study, we found that *Nl*Gr23a is required for the repulsive response to OA in the BPH.

## Figures and Tables

**Figure 1 cells-12-00771-f001:**
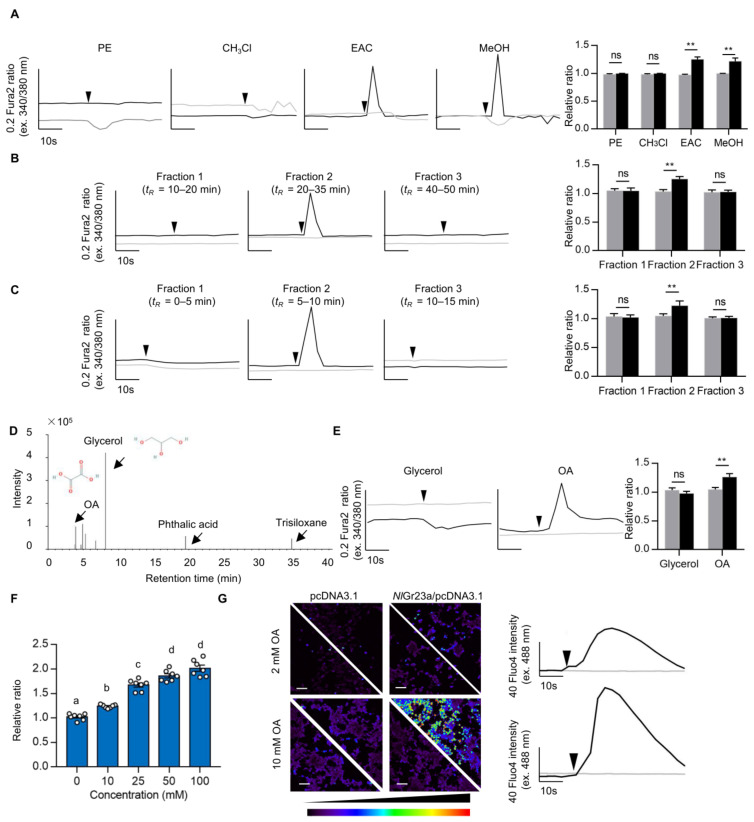
Oxalic acid is a ligand of *Nl*Gr23a. (**A**) Round 1 of ligand screening. The responses of *Sf9* cells transfected with pIZ-*Nl*Gr23a-V5-His (black lines) or pIZ-V5-His (grey lines) vector to different fractions from the crude extract of rice plants separated by petroleum ether (PE), chloroform (CHCl_3_), ethyl acetate (EAC), and methyl alcohol (MeOH) were analyzed by calcium imaging. The changes in intracellular Ca^2+^ are indicated by the ratio of F_340_/F_380_ in Fura2-AM loaded cells. Each curve represents one cell responding to a specific fraction. The vertical scale bar indicates 0.2 ratio of Fura2 fluorescence intensity excited at 340 and 380 nm, and the horizontal scale bar indicates 10 s. Arrowheads indicate timing of stimulation. The rightmost chart shows Ca^2+^ response of *Sf9* cells transfected with pIZ-*Nl*Gr23a-V5-His (black columns) or pIZ-V5-His (grey columns). Ca^2+^ response is indicated by the Fura2 ratio normalized to the baseline before stimulation. Bars represent means ± SEM. Mann–Whitney non parametric test; ns means no statistical difference (*p* > 0.05); ** *p* < 0.01 (*n* ≥ 7). (**B**) Round 2 of ligand screening. Representative traces evoked by three semi-preparative high-performance liquid chromatographic (semi-prep-HPLC) fractions separated from the EAC fraction. See Appendix A for HPLC chromatogram of the EAC fraction. (**C**) Round 3 of ligand screening. Representative traces evoked by three semi-prep-HPLC fractions separated from the bioassay-positive fraction from round 2 of ligand screening. See Appendix A for HPLC chromatogram of the bioassay-positive subfraction. (**D**) GC-MS chromatogram of bioassay-positive fraction from round 3 of ligand screening. Arrowheads indicate substances identified in the bioassay-positive fraction. See Appendix A for GC-MS chromatogram of the control solution and Appendix A for the retention time and response intensity of each compound. Chemical structure of oxalic acid (OA, PubChem CID: 971; date last accessed: Feb 2023) and glycerol (PubChem CID: 753; date last accessed: Feb 2023) were obtained from *PubChem* (https://pubchem.ncbi.nlm.nih.gov). (**E**) Round 4 of ligand screening. Representative traces evoked by 5 mM glycerol and 5 mM OA. (**F**) Ca^2+^ response of the *Nl*Gr23a-*Sf9* cells stimulated with various concentrations of OA solutions. Ca^2+^ response is indicated by Fura2 ratio. The ratio is normalized to the baseline before stimulation. Bars represent means ± SEM. Different letters (a to d) above bars represent significantly different groups. Multiple samples were analysed using ANOVA at *p* < 0.05 followed by Duncan’s multiple range test (*n* = 7). (**G**) Ca^2+^ responses before and after OA application in HEK293T cells. Divided panels of negative controls, transfected with pcDNA3.1 (left), or the *Nl*Gr23a/pcDNA3.1 transfected HEK293T cells (right), before and after application of either 2 mM (upper) or 10 mM OA (lower). White scale bar indicates 100  µm. The color scale indicates the fluorescent intensity. The rightmost curves represent representative traces evoked by OA in HEK293T cells transfected with *Nl*Gr23a/pcDNA3.1 (black lines) or pcDNA3.1 (grey lines). The changes in intracellular Ca^2+^ were indicated by Fluo4 fluorescence intensity. The vertical scale bar indicates 40 Fluo4 fluorescence intensity excited at 488 nm, and the horizontal scale bar indicates 10 s. Arrowheads indicate timing of stimulation.

**Figure 2 cells-12-00771-f002:**
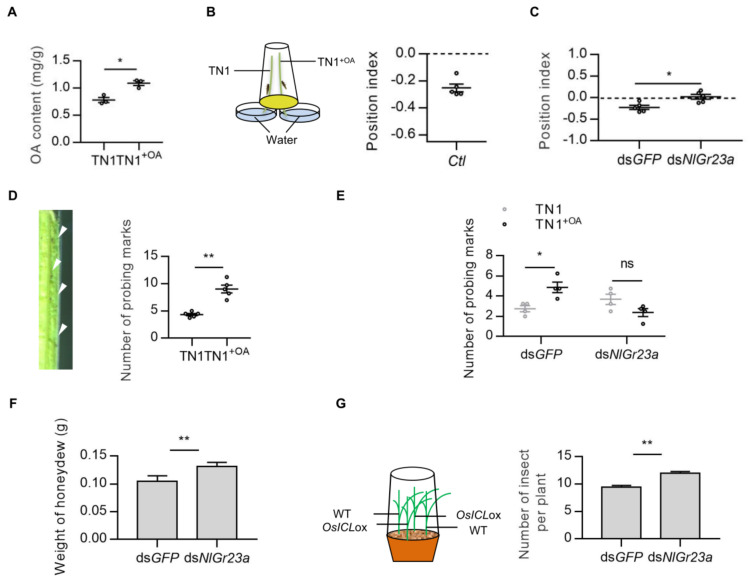
*NlGr23a* mediated the antifeedant activity of oxalic acid in rice plants. (**A**) OA content before and after immersion treatment in TN1. The TN1^+OA^ represents TN1 treated with 0.2% OA solution. Bars represent means ± SEM. Mann–Whitney non-parametric test; ** p* < 0.05 (*n* = 3). (**B**) Left: Schematic representation of the experimental apparatus used for dual-choice test with rice plants. Right: Collective data (*n* = 5) of the position index (PI) of *Ctl* in response to TN1^+OA^. Bars represent means ± SEM. (**C**) PI values for *NlGr23a* dsRNA-treated and *GFP* dsRNA-treated BPHs in response to TN1^+OA^ in dual choice assays. Bars represent means ± SEM. Mann–Whitney non-parametric test; ** p* < 0.05 (*n* = 5). (**D**) Left: Probing marks left on the surface of a rice stem. Right: Collective data (*n* = 5) of the number of probing marks on the surface of TN1^+OA^ or TN1 after BPH infestation. Bars represent means ± SEM. Mann–Whitney non-parametric test; *** p* < 0.01. (**E**) Collective data (*n* = 4) of the number of probing marks on the surface of TN1^+OA^ or TN1 made by BPHs treated with *NlGr23a* dsRNA or *GFP* dsRNA. Bars represent means ± SEM. Mann–Whitney non-parametric test; ns means no statistical difference (*p* > 0.05); ** p* < 0.05. (**F**) Weight of honeydew secreted by three female BPH adults after 48 h feeding on *OsICL*-overexpressing transgenic plants (*OsICL*ox). Bars represent means ± SEM. Mann–Whitney non-parametric test; *** p* < 0.01 (*n* = 12). (**G**) Left: Schematic representation of the experimental apparatus used for host choice test. Right: Number of female BPH adults per *OsICL*ox plant at 48 h post-infestation. Bars represent means ± SEM. Mann–Whitney non-parametric test; *** p* < 0.01 (*n* = 12).

**Figure 3 cells-12-00771-f003:**
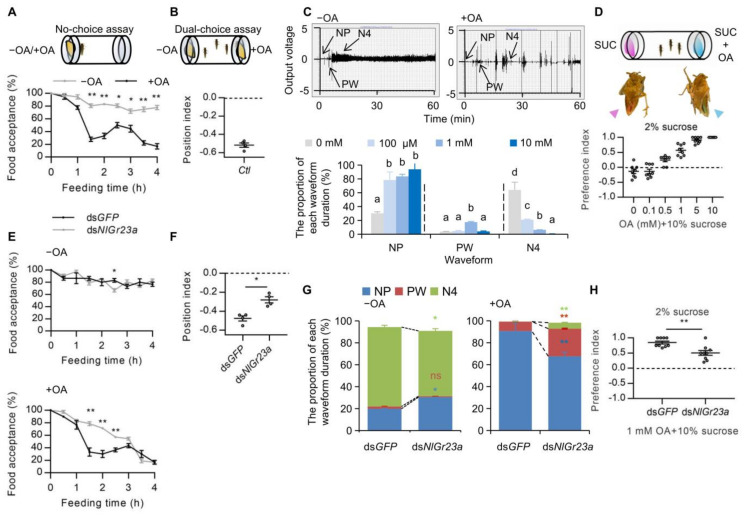
*NlGr23a* mediated the antifeedant activity of oxalic acid in artificial diets. (**A**) Top: Schematic representation of the experimental apparatus used for no-choice test. Bottom: The proportion of BPHs accepting artificial diets without (−OA, in grey) or with 100 µM OA (+OA, in black) as food in no-choice test. Bars represent means ± SEM. Mann–Whitney non-parametric test; ** p* < 0.05; *** p* < 0.01 (*n* = 3). (**B**) Top: Schematic representation of the experimental apparatus used for dual-choice test. Bottom: Collective data (*n* = 4) of the PI values of the laboratory insects (*Ctl*) in response to food containing 100 µM OA in dual-choice assays. Bars represent means ± SEM. (**C**) Top: Overall typical view of electrical penetration graph (EPG) waveforms generated by the feeding behaviors of the BPH on liquid diet sacs (LDS) without (left) or with (right) 100 µM OA. There are three typical waveforms, including non-penetration (NP), pathway (PW), and ingestion (N4), which are indicted by arrows. Bottom: The duration of each EPG waveform as a proportion of observation time produced by BPHs on LDS containing artificial diets with different concentrations of OA. Bars represent means ± SEM. Different lowercase letters above columns represent significant differences at *p* < 0.05 from Duncan’s multiple range test (*n* = 5). (**D**) Dose-responsive food choice assay for OA. Top: Schematic representation of the experimental apparatus used for food choice assay. Starved BPHs were introduced into a glass tube and provided with two choices (sucrose versus sucrose plus OA) colored with either red or blue dye. Bottom: Concentration-dependent avoidance of OA in BPHs. Bars represent means ± SEM (*n* = 8). (**E**) Top: The proportion of insects accepting artificial food absent of OA (−OA) among ds*GFP*-treated (black line) or ds*NlGr23a-*treated (grey line) BPHs in no-choice tests. Bottom: The proportion of insects accepting artificial food with 100 µM OA (+OA) among ds*GFP*-treated or ds*NlGr23a-*treated BPHs in no-choice tests. Bars represent means ± SEM. Mann-Whitney non parametric test; ** p* < 0.05; *** p* < 0.01 (*n* = 3). (**F**) PI values of ds*NlGr23a*-treated and ds*GFP*-treated BPHs in response to +OA diets in dual-choice assays. Bars represent means ± SEM. Mann–Whitney non-parametric test; ** p* < 0.05 (*n* = 4). (**G**) The duration of each EPG waveform as a proportion of observation time produced by ds*GFP*-treated and ds*NlGr23a*-treated BPHs on LDS with or without 10 mM OA. Stacked bar graphs represent means ± SEM. Mann–Whitney non-parametric test; ns means no statistical difference (*p* >0.05); ** p* < 0.05; *** p* < 0.01 (*n* = 5). (**H**) Food choice assay for ds*NlGr23a*-treated and ds*GFP*-treated BPHs. 1 mM OA was mixed with 10% sucrose. Bars represent means ± SEM. Mann–Whitney non-parametric test; *** p* < 0.01 (*n* = 9).

**Figure 4 cells-12-00771-f004:**
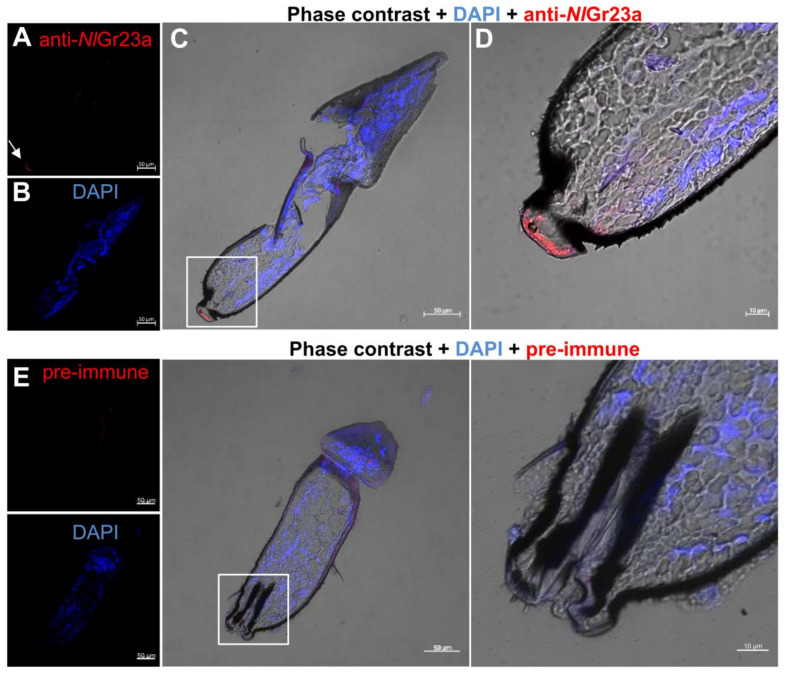
Localization of *Nl*Gr23a in the distal part of the female labium of the BPH. The labial cryosections were stained with rabbit anti-*Nl*Gr23a antiserum in combination with Alexa Fluor 555^®^-conjugated goat anti-rabbit IgG. Fluorescence images of anti-*Nl*Gr23a (**A**), DAPI (**B**), the merged fluorescence image (**C**), and the higher magnification image of the merged image (**D**) are shown. Scale bar represents 50 µm (**A**–**C**) or 10 µm (**D**). Arrowhead points to the cell bodies stained by anti-*Nl*Gr23a antiserum. (**E**) Negative control experiments were conducted by application of pre-immune rabbit serum taken from the animal immunized with *Nl*Gr23a peptide to produce the primary antiserum, together with Alexa Fluor 555^®^-conjugated goat anti-rabbit IgG; no immunoreactive cells were observed in the labial tip. Scale bars represent 50 µm, except for the magnified image, in which the scale bar represents 10 µm.

## Data Availability

All data generated or analyzed during this study are included in this published article and its Appendix A.

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
