# Peer review of "Oxalic Acid Inhibits Feeding Behavior of the Brown Planthopper via Binding to Gustatory Receptor Gr23a"

_cells, 2023, doi:10.3390/cells12050771_

Round 1

Reviewer 1 Report

This manuscript identified oxalic acid (OA) as a ligand of NlGr23a by the heterologous expression systems, and also showed that NlGr23a is essential for OA’s antifeedant activity in rice plants and artificial diets. These results on rice-planthopper interactions will be of broad interest for pest control. However, before potential acceptance, some questions should be addressed.

Comments:

1.     Can the title be changed to “Oxalic Acid Inhibits Feeding behavior of the Brown Planthopper via Binding to Gustatory Receptor Gr23a”?

2.     Sometimes “behavior”, sometimes “behaviour”, please keep the same.

3.     L48, the brown planthopper first appears, please give the Latin name.

4.     L82, TN1 first appears, please give the full name.

5.     L104, by using >> as

6.     L106, foetal >> fetal

7.     For the mammalian HEK293T cell line, why not obtain stable cell lines of NlGr23a under G418 selection?

8.     L119-120, how many microliters Lipo3000 are used?

9.     In Section 2.3.1, sf9 cell transfection used Fugene HD transfection reagent, but in Section 2.5, sf9 cell transfection used Lipo3000, why?

10.  L186, GFP should be italic; L202, β-actin should be italic.

11.  Section 2.8 is confusing, please give more information.

12.  Figure 2F, Y title is wrong; X title, WT is dsGFP or OsICLox is dsNlGr23a?

13.  Figure 2G, Y title, plan >> plant, Number of, sometimes “NO. of”, please keep the same.

14.  The legend of Figure 2G can be changed to “Number of dsNlGr23a-treated female BPH adults on each plant”

15.  For EPG waveforms, there are np, N1, N2, N3, N4-a, N4-b, and N5, why only choose three types?

16.  Figure 3C should be rearranged.

17.  Figure 3D, statistical analyses?

18.  L472, L486, means + SEM >> means ± SEM

19.  L524, “Such factors may also explain why”, why are italic?

20.  L571, Sf9 is Spodoptera frugiperda?

21.  L574, qRT-PCR is Quantitative reverse transcription PCR?

22.  Table S1, the primers for GFP?

23.  Reference style does not meet the requirements of the journal.

Reviewer 2 Report

 In this study, Kang et al. revealed that oxalic acid (OA), one of the secondary compounds produced by plants, is a ligand for NlGr23a, a gustatory receptor (Gr) of the brown planthopper. They also found that NlGr23a is required for the repulsive response of the brown planthopper to OA in rice. These results are interesting in that they provide a new mechanism for GR-mediated repulsive responses of insects.  However, the data showing that OA is a ligand for NlGr23a needs to be more convincing, and the following issues need to be addressed.

Major

1.     For fig. 1A~C, and E, please indicate the timing of ligand application and enlarge the figure so that the shape of the peak is clear.  The trace after 60 seconds can be omitted. In addition to representative trace data, quantitative data showing the difference between mock-transfected and NlGr23a-transfected cells is needed.

2.   Many dots appeared in the lower right panel of fig. 1G looks somewhat different from normal Ca response seen in healthy HEK293T cells, raising the concern cells just became unhealthy upon ligand stimulation, so it is necessary to clarify whether this is a normal response. Please increase the resolution of the figure to show cells not only at the peak of the response, but also after the response has disappeared. Also, please display the trace data as in fig. 1E.

3.     In my opinion, microscale thermophoresis experiments using lysates of cells expressing the receptor of interest do not seem convincing to verify the interaction with the ligand, since the conformation of the receptor may not be maintained and various other proteins are present. Please explain the certainty of this approach and show the results of a control experiment using mock-transfected cells in addition to fig. S4A.

4.     In fig. 4, the localization of NlGr23a in the labium is somewhat convincing, but the high background signal of the pre-immune serum seen in fig. S9 raises concerns about the reliability of the conclusion. Please show convincing data that includes negative control data in the same figure.

5.     In the introduction, it would be good to have an explanation of why the authors focused on NiGr23a among many other GRs.

Minor

1.     In figures 2F and G, the labels do not match the descriptions in the text.

2.     In the vertical labels in fig. 2G, could "plan" be a mistake for "plant" ?

3.     In line 418, "essential" is overstated because the repulsive responses to OA are still present in dsNiGr23a animals in their experiments.

Round 2

Reviewer 1 Report

Minor revisions:

1.     L591, Sf9 >> Spodoptera frugiperda 9 cells

2.     L594, qRT-PCR >> Quantitative real-time PCR

Author Response

Point 1: L591, Sf9>>Spodoptera frugiperda 9 cells

Response 1: The text has been modified. Please see L561.

Point 2: L594, qRT-PCR >> Quantitative real-time PCR

Response 2: The text has been modified. Please see L589.

Reviewer 2 Report

The authors have adequately addressed my comments. If only the following point is addressed, I think this manuscript can be published.

As the authors suggested, Fig. S4A, which lacks control data, should be deleted.

Author Response

Point 1: As the authors suggested, Fig. S4A, which lacks control data, should be deleted.

Response 1: Fig. S4A has been deleted. The corresponding sentences in the Results, Fig. S4 and References have been updated accordingly.